# CMMD: Contrastive Multi-Modal Diffusion for Video-Audio Conditional Modeling

## Abstract

We introduce a multi-modal diffusion model tailored for the bi-directional conditional generation of video and audio. We propose a joint contrastive training loss to improve the synchronization between visual and auditory occurrences. We present experiments on multiple datasets to thoroughly evaluate the efficacy of our proposed model. The assessment of generation quality and alignment performance is carried out from various angles, encompassing both objective and subjective metrics. Our findings demonstrate that the proposed model outperforms the baseline in terms of quality and generation speed through introduction of our novel cross-modal easy fusion architectural block. Furthermore, the incorporation of the contrastive loss results in improvements in audio-visual alignment, particularly in the high-correlation video-to-audio generation task.

## 1 Introduction

Multi-media generation with diffusion models has attracted extensive attention recently. Following breakthroughs in image (Ramesh et al., 2022) and audio generation (Liu et al., 2023), multi-media generation like video remains challenging due to increased data and content size and the added complexity of dealing with both audio and visual components. Challenges for generating multi-modal content include 1) time variant feature maps leading to computationally expensive architecture and 2) audio and video having to be coherent and synchronized in terms of semantics and temporal alignment.

Existing research has predominantly concentrated on unidirectional cross-modal generation, such as producing audio from video cues (Luo et al., 2023; Zhu et al., 2023) and vice versa (Jeong et al., 2023; Lee et al., 2023). These approaches typically employ a conditional diffusion model to learn a conditional data distribution $p(x|y)$. Although these models have shown considerable promise, their unidirectional nature is a limitation; a model trained for $p(x|y)$ is not suited for tasks requiring $p(y|x)$. However, Bayes' theorem elucidates that a joint distribution can be decomposed into $p(x, y) = p(x|y)p(y) = p(y|x)p(x)$, suggesting that the construction of a joint distribution inherently encompasses bi-directional conditional distributions. With the advent of the iterative sampling procedure in diffusion models, classifier guidance (Dhariwal & Nichol, 2021; Song et al., 2021b; Ho et al., 2022) has emerged as a viable approach for training an unconditional model capable of conditional generation. This approach has been extensively adopted in addressing the inverse problems associated with diffusion models, such as image restoration (Kawar et al., 2022) and text-driven generation (Ramesh et al., 2021).

MM-diffusion (Ruan et al., 2023) represents a groundbreaking foray into the simultaneous modeling of video and audio content. The architecture employs a dual U-Net structure, interconnected through cross-attention mechanisms (Vaswani et al., 2017), to handle both video and audio signals. Although MM-diffusion demonstrates impressive results in terms of *unconditional* generation quality, it has two major limitations: Firstly, it's random-shift cross-attention mechanism is still complex and it relies on a super-resolution upscaling model to improve image quality. Secondly, the focus has been on unconditional generation, while we focus on conditional generation and improve the evaluation methodology.

In this study, we introduce an improved multi-modal diffusion architecture with focus on bi-directional *conditional* generation of video and audio. This model incorporates an optimized design that more effectively

integrates video and audio data for conditional generation tasks. Furthermore, we leverage a novel joint contrastive diffusion loss to improve alignment between video and audio pairs. Our experiments on two different dataset employ both subjective and objective evaluation criteria. We achieve superior quality than the baseline and stronger synchronization.

The key contributions can be summarized as follows:

- We present an optimized version of the multi-modal *latent-spectrogram* diffusion model, featuring a pretrained video autoencoder, a vocoder and an easy fusion mechanism. This design aims to more effectively integrate cross-modality information between video and audio, while also enhancing *conditional* sampling quality.

- Drawing inspiration from uni-modal contrastive learning, we propose a novel contrastive loss function tailored for the joint model. This function is instrumental in enhancing the alignment accuracy for the conditional generation of video-audio pairs.

- Our extensive experimental evaluations, performed on two distinct datasets, AIST++ (Li et al., 2021) and EPIC-Sound (Huh et al., 2023), cover a variety of video-audio scenarios. We propose to use metrics with improved correlation human perception and practical relevance compared to prior work in the field, as we find several widely used metrics to have strong deficiencies. The assessments, based on a range of subjective and objective metrics demonstrate that our method outperforms the existing MM-diffusion (Ruan et al., 2023) in terms of quality, as well as non-contrastive variants in terms of temporal synchronization.

## 2 Related Work

**Diffusion Models:** Demonstrating remarkable efficacy in image generation tasks, probabilistic diffusion models have emerged as a robust alternative to highly-optimized Generative Adversarial Networks (GANs) (Goodfellow et al., 2014). The superior performance of diffusion models is attributed to their stability during the training process (Song & Ermon, 2019; Ho et al., 2020; Song et al., 2021b; Song & Ermon, 2019; Kingma et al., 2021; Dhariwal & Nichol, 2021). These models typically employ a parameter-free diffusion process that degrades the original signal, followed by a denoising process using a trained U-Net like architecture to restore the signal. The optimization objective of the diffusion model can be derived from either the variational inference or stochastic differential equation perspectives (Ho et al., 2020; Song et al., 2021b). Beyond image generation, there has been increasing interest in leveraging diffusion models for conditional generation tasks, including image compression, translation, and enhancement (Saharia et al., 2022b; Preechakul et al., 2022; Yang & Mandt, 2022; Saharia et al., 2022a). Recent advancements in diffusion models (Ramesh et al., 2022; Rombach et al., 2022) incorporate the diffusion-denoising process in the latent space, aiming to enhance the scalability of diffusion models for high-resolution images. In light of their impressive results in image-related tasks, it is a logical progression to extend the application of these models to video and audio signals (Ho et al., 2022; Yang et al., 2022; Blattmann et al., 2023; Voleti et al., 2022; Kong et al., 2020b; Zhang et al., 2023). To learn the joint distribution of a sequence, these models are further refined to account for the temporal coherence of the signals.

**Advancements in Video-Audio Cross-Modality Models:** The domain of deep generative models has recently experienced a significant uptick in interest, particularly in the area of cross-modal generation—an application that is currently undergoing rapid evolution. Historically, the majority of research in this field has been primarily focused on text-to-visual tasks, as evidenced by various studies (Li et al., 2019; Singer et al., 2022; Gafni et al., 2022; Zhang et al., 2023). However, a discernible trend towards the more intricate audio-video modality has emerged (Lee et al., 2022; Ge et al., 2022; Di et al., 2021), driven by the potential to create more innovative and engaging content within this sphere. Concurrently, diffusion models have begun to assume a pivotal role in related research. These models, with their ability to model complex distributions, have found a natural application in the cross-modal generation task. For example, TPoS (Jeong et al., 2023) and Soundini (Lee et al., 2023) are two recent models that demonstrate proficiency in audio-to-video generation. Their success underscores the potential of diffusion models in this domain. Other models

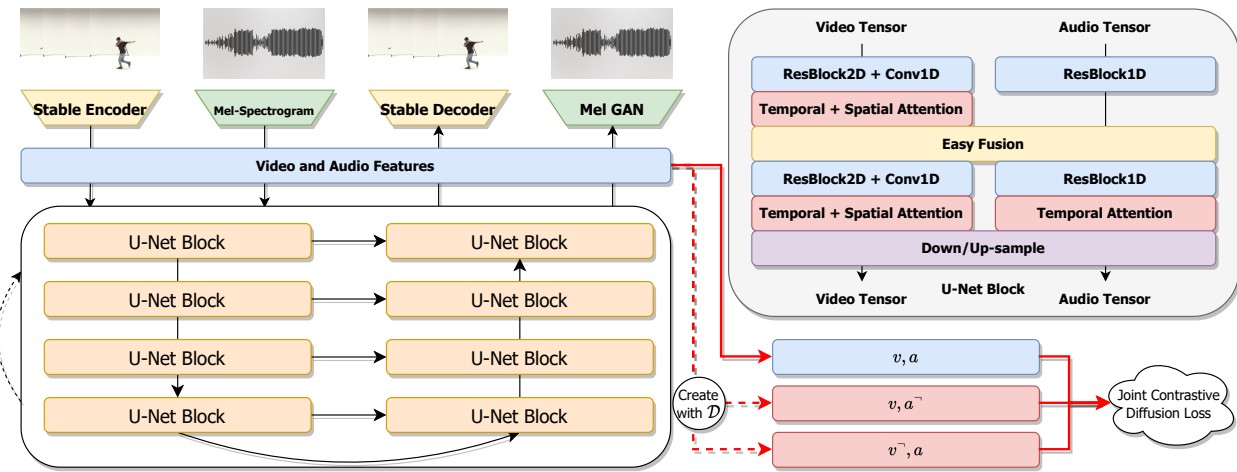

Figure 1: Overview of our proposed architecture and method. The detailed implementation of each U-Net block is depicted in the upper right corner. Training of the diffusion model is performed on latent-spectrogram space.

such as CDCD (Zhu et al., 2023) and Diff-Foley (Luo et al., 2023) specifically target the video-to-audio problem. These models represent a growing interest in reverse modality tasks, expanding the boundaries of what is possible in the realm of cross-modal generation. Notably, MM-diffusion (Ruan et al., 2023), which emphasizes the simultaneous generation of both video and audio, is, to our knowledge, the first model capable of managing both video-to-audio and audio-to-video generation. Despite its impressive performance in low-resolution unconditional generation, its computational efficiency and conditional generation performance warrant further investigation.

## 3 Method

In this section, we provide an overview of the diffusion model employed, followed by a description of the intricacies of the architecture design of the proposed model. Finally, we introduce the joint contrastive loss that enhances the alignment of video and audio components. An overview of our model is shown in Fig. 1.

### 3.1 Video-Audio Joint Diffusion Model

**Diffusion models** represent a category of hierarchical latent variable models used for data generation through a series of iterative stochastic denoising steps (Sohl-Dickstein et al., 2015; Ho et al., 2020; Song et al., 2021a; Song & Ermon, 2019). These models establish a joint distribution encompassing both the original data, denoted as $\mathbf{x}_0$, and its perturbed variants $\mathbf{x}_{1:N}$. In this framework, there are two key processes at play: the diffusion process, denoted as $q$, which progressively erases structural information, and its counterpart, $p_\theta$, which regenerates the structure. These processes involve Markovian dynamics across a sequence of transitional steps (Ho et al., 2020), symbolized as $n$ and can be described using the following equations:

$$q(\mathbf{x}_n|\mathbf{x}_{n-1}) = \mathcal{N}(\mathbf{x}_n|\sqrt{1 - \beta_n}\mathbf{x}_{n-1}, \beta_n\mathbf{I});$$
$$p_\theta(\mathbf{x}_{n-1}|\mathbf{x}_n) = \mathcal{N}(\mathbf{x}_{n-1}|\mu_\theta(\mathbf{x}_n, n), \beta_n\mathbf{I}). \tag{1}$$

The variance, denoted as $\beta_n \in (0, 1)$, typically adheres to a predetermined schedule, such as linear or cosine scheduling (Nichol & Dhariwal, 2021). Notably, the diffusion process is parameter-free, while the denoising process relies on a neural network parameterized by $\theta$ to predict the posterior mean.

Denoising diffusion models introduced a practical objective function for training the reverse process (Ho et al., 2020; Salimans & Ho, 2022; Hang et al., 2023):

$$L(\theta, \mathbf{x}_0) = \mathbb{E}_{n,\epsilon} \left[ w(n)||\mathbf{x}_0 - \mathbf{x}_\theta(\mathbf{x}_n(\mathbf{x}_0), n)||^2 \right] \tag{2}$$

where $n$ follows a uniform distribution $\text{Unif}\{1, ..., N\}$, $\epsilon$ is sampled from a standard normal distribution, $\mathbf{x}_n(\mathbf{x}_0) = \sqrt{\alpha_n}\mathbf{x}_0 + \sqrt{1 - \alpha_n}\epsilon$, $\mathbf{x}_\theta(\cdot)$ reconstruct $\mathbf{x}_0$ and $\alpha_n = \prod_{i=1}^{n}(1 - \beta_i)$. This formula characterizes a universal diffusion model loss with an adjustable weighting term, $w(n)$, which connects various parameterizations of the prediction model $\theta$:

$$
\begin{aligned}
\text{Eq.2} &\equiv \mathbb{E}_{n,\epsilon}[||\epsilon - \epsilon_\theta(\mathbf{x}_n(\mathbf{x}_0), n)||^2], \quad \text{when } w(n) = \frac{\alpha_n}{1 - \alpha_n} \\
&\text{or } \mathbb{E}_{n,\epsilon}[||\mathbf{v} - \mathbf{v}_\theta(\mathbf{x}_n(\mathbf{x}_0), n)||^2], \quad \text{when } w(n) = \frac{1}{1 - \alpha_n}
\end{aligned}
\tag{3}
$$

where $\epsilon_\theta$ represents the most commonly used parameterization in previous works (Karras et al., 2022; Ho et al., 2020; Ruan et al., 2023; Dhariwal & Nichol, 2021; Rombach et al., 2022; Song et al., 2021a), and $\mathbf{v}_\theta$ (velocity) has also shown promising results with a more stable training process (Salimans & Ho, 2022). We adopt the latter method to train our model.

**Video-Audio Modeling** Our approach to video-audio joint modeling follows a design analogous to the uni-modal diffusion model. Here, the data point $\mathbf{x}$ comprises two modalities: the video signal $v_{0..N}$ and audio signal $a_{0..N}$. Consequently, the optimization objective resembles the form in Eq.3:

$$
\mathcal{L}_{\text{jdiff}} = \mathbb{E}_{n,\epsilon}[||\mathbf{v} - \mathbf{v}_\theta(v_n, a_n, n)||^2]
\tag{4}
$$

where $\mathbf{v}$ represents the velocity parameterization for both video and audio, specifically $\mathbf{v} = [\sqrt{\alpha_n}\epsilon_v - \sqrt{1 - \alpha_n}v_0, \sqrt{\alpha_n}\epsilon_a - \sqrt{1 - \alpha_n}a_0]$. This implies that the model $\mathbf{v}_\theta$ simultaneously predicts two outputs, embodying a joint diffusion model that effectively manages both modalities.

**Guided Conditional Generation** An intriguing aspect of diffusion models is their capacity to enable conditional generation through guidance from a classifier, even in the context of models trained without conditioning (Dhariwal & Nichol, 2021). Typically, this guidance method involves an additional classifier, $p_\phi(y|x)$, and utilizes the gradient term $\nabla_x p_\phi(y|x)$ to adjust the sampling direction during the denoising process.

However, in our model, which considers both video and audio modalities, we can employ a more straightforward *reconstruction guidance* approach (Ho et al., 2022). For the video-to-audio generation case, we can formalize conditional generation as follows (audio-to-video shares a similar formulation).

$$
\begin{aligned}
&1. \ v_n = \sqrt{\alpha_n}v_0 + \sqrt{1 - \alpha_n}\epsilon_v \text{ where } \epsilon_{v,a} \sim \mathcal{N}(0, I) \\
&2. \ \mathbf{v}_v, \mathbf{v}_a = \mathbf{v}_\theta(v_n, a_n, n) \\
&3. \ \hat{v}_0 = \frac{\sqrt{\alpha_n}\epsilon_v - \mathbf{v}_v}{\sqrt{1 - \alpha_n}}, \bar{a}_0 = \frac{\sqrt{\alpha_n}\epsilon_a - \mathbf{v}_a}{\sqrt{1 - \alpha_n}} \\
&4. \ \hat{a}_0 = \bar{a}_0 - \underbrace{\lambda\sqrt{\alpha_n}\nabla_{a_n}||v_0 - \hat{v}_0||^2}_{\text{reconstruction guidance}} \\
&5. \ a_{n-1} = \sqrt{\alpha_{n-1}}\hat{a}_0 + \sqrt{1 - \alpha_{n-1}}\epsilon_a
\end{aligned}
\tag{5}
$$

where the gradient guidance is weighted by $\lambda$, and in the case of $\lambda = 0$, the generation scheme is equivalent to the *replacement* method (Song et al., 2021b). Both $\epsilon_a$ and $\epsilon_v$ are drawn from an isotropic Gaussian prior at the start of the iteration. Therefore, these equations depict an intermediate stage of the conditional generation process using the DDIM sampling method (Song et al., 2021a). Although the speed of sampling is not the primary focus of our model, alternative ODE or SDE solvers can be employed to expedite the denoising sampling process (Lu et al., 2022; Karras et al., 2022).

## 3.2 Architecture

Like most previous diffusion models, our architectural framework adheres to the U-Net-based design paradigm (Ho et al., 2020; 2022; Ruan et al., 2023; Rombach et al., 2022; Ronneberger et al., 2015). To

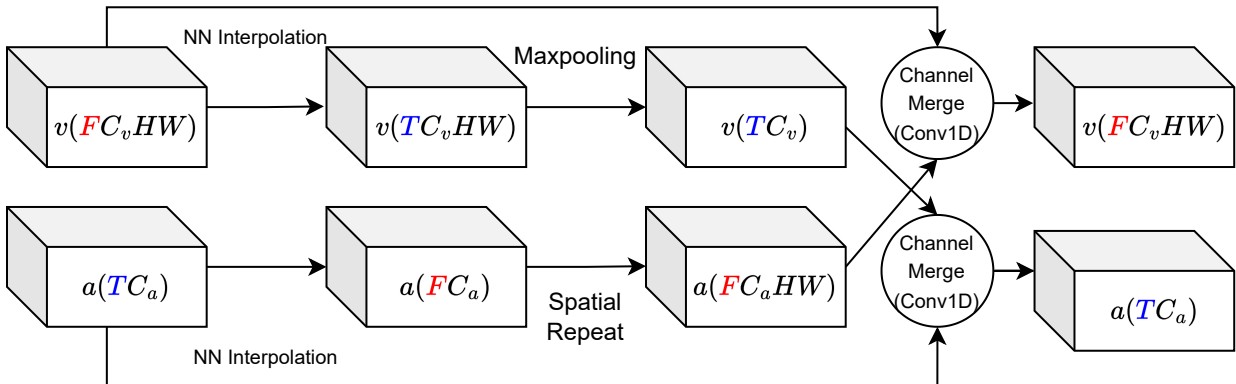

Figure 2: Easy Fusion: For brevity, we use the symbol $v$ to denote video tensors, where $F$ (Frames) $\times$ $C_v$ (Channels) $\times$ $H$ (Height) $\times$ $W$ (Width) represents the shape of the tensor. Similarly, we use the symbol $a$ to represent audio tensors, with $T$ (Timesteps) $\times$ $C_a$ (Channels) denoting the shape of the audio tensor. $v$ and $a$ tensors are concurrently processed and merged in the final step. NN interpolation represents nearest neighbor interpolation.

effectively process signals originating from dual modalities with distinct dimensionalities, we employ a combination of 2D+1D Residual blocks with Temporal-Spatial attentions for video inputs. For audio inputs, we only use 1D Residual blocks with Temporal Attention. Although the foundational architecture is partly inspired by MM-diffusion (Ruan et al., 2023), which utilizes dual interconnected U-Nets for processing audio and video separately, our approach incorporates a more efficient feature fusion mechanism specifically designed to meet the demands of conditional generation.

**Latent-Spectrogram Diffusion**  The training and evaluation of a multi-modal diffusion model can pose significant computational challenges. To address this issue, we adopt a methodology akin to latent diffusion (Rombach et al., 2022) for the purpose of reducing the feature dimensionality. In particular, we employ a pre-trained autoencoder to compress video frames into a concise representation while minimizing the loss of semantic information. This approach not only enables our model to accommodate higher video resolutions within GPU memory limitations but also enhances its ability to capture temporal dependencies in videos (Blattmann et al., 2023). For audio signals, we use a time-frequency representation, specifically the Mel-spectrogram. This transformation yields a more compact representation with frequency channels that closely align with human auditory perception.

**Improved MelGAN Vocoder**  The conversion of the generated Mel spectrogram back to an audio waveform was accomplished by training a vocoder on the MelGAN (Kumar et al., 2019) architecture. We incorporated several improvements from Hifi-GAN (Kong et al., 2020a), such as transitioning to a least-squares GAN loss and adjusting the loss weightings, with a particular emphasis on the Mel-spectrogram loss.

**Easy Fusion and Implicit Cross-Attention**  Our model's capacity to handle inputs and outputs from two distinct modalities presents a considerable challenge in terms of aligning feature maps and merging semantic information for cross-modal conditioning. While conventional cross-attention mechanisms (Vaswani et al., 2017) offer an approach to bridging these signals, they can become computationally inefficient as the length of time sequences increases. In contrast, MM-Diffusion (Ruan et al., 2023) resorts to randomly truncating time windows to alleviate the computational load, yet this method inevitably results in a loss of receptive field.

In response to this challenge, we introduce our computationally more efficient *easy fusion* method, illustrated in Fig. 2. This design includes nearest neighbor, pooling, and repetition, to guarantee that both video and audio tensors maintain consistent temporal/spatial shapes, enabling their concatenation along the channel dimension. Another crucial consideration is that the most recent U-Net designs for diffusion models incorpo-

rate a self-attention module (Ruan et al., 2023; Rombach et al., 2022; Ho et al., 2022), offering the potential to alleviate computational overhead from cross-attention. Within the context of easy fusion, we assume that the attention module can be described by:

$$\text{CrossAttention}(v, a) \cong \text{SelfAttention}(\text{EasyFusion}(v, a)). \tag{6}$$

We posit that the self-attention operating in this manner implicitly signifies the existence of an inherent cross-attention mechanism, thereby rendering an additional attention block obsolete.

### 3.3 Joint Contrastive Training

To improve the synchronization of video and audio in our model, we utilize principles of contrastive learning (Oord et al., 2018). This approach has proven effective in maximizing the mutual information $I(a; v)$ for video-to-audio conditional generation (Zhu et al., 2023; Luo et al., 2023). The CDCD (Zhu et al., 2023) method seamlessly integrates contrastive loss into the video-to-audio conditional diffusion models, as given by

$$\begin{aligned}
\mathcal{L}_{\text{cont}} &:= \mathbb{E}_A \log \left[ 1 + \frac{p_\theta(a_{0:N})}{q(a_{0:N}|v_0)} M \mathbb{E}_{A'} \left[ \frac{p_\theta(a_{0:N}^-|v_0)}{q(a_{0:N}^-)} \right] \right] \\
&\approx \mathcal{L}_{\text{cdiff}}(a_{0:N}, v_0) - \eta \sum_{a_0^- \in A'} \mathcal{L}_{\text{cdiff}}(a_{0:N}^-, v_0)
\end{aligned} \tag{7}$$

where the set $A$ includes the correct corresponding audio samples, while $A'$ contains the mismatched negative samples of $A$. $L_{\text{cdiff}}$ denotes the unimodal conditional diffusion loss, with $v$ representing the conditioning videos and $M$ indicating the number of negative samples. To streamline the training process, we replace $M$ with a weighting term $\eta$, eliminating the need to generate $M$ negative samples at each training step. This means at each training step, we can sub-sample a batch of $a^-$ from the $M$ samples for computational efficiency.

The above formulation pertains to training a classifier-free conditional diffusion model. To adapt this approach to our joint diffusion loss, as described in Eq.4, we observe that we are training an implicit conditional diffusion model $p_\theta(a_{n-1}|a_n, v_n)$. Eq.5 demonstrates that $v_n$ can be directly calculated during conditional generation:

$$v_n \sim q(v_n|v_0) = \sqrt{\alpha_n} v_0 + \sqrt{1 - \alpha_n} \epsilon_v \tag{8}$$

which implies that $v_{1:N}$ is fixed with a given $\epsilon_v$ and $v_0$. Given this relationship between $v_n$ and $v_0$, we have following approximation $p_\theta(a_{n-1}|a_n, v_0) \approx p_\theta(a_{n-1}|a_n, v_n)q(v_n|v_N)$, where $v_N$ denotes the sampled noise $\epsilon_v$. Thus, we can bridge Eq.7 to our jointly trained multi-modal diffusion model. For audio-to-video generation, we can follow the same method above by swapping $v$ and $a$. Finally, the resulting joint contrastive loss can be represented by the following three terms:

$$\begin{aligned}
\mathcal{L}_{\text{cont}} = \mathcal{L}_{\text{jdiff}}(a_{0:N}, v_{0:N}) &- \eta \mathbb{E}_{a_0^- \sim A'} \mathcal{L}_{\text{jdiff}}(a_{0:N}^-, v_{0:N}) \\
&- \eta \mathbb{E}_{v_0^- \sim V'} \mathcal{L}_{\text{jdiff}}(a_{0:N}, v_{0:N}^-)
\end{aligned} \tag{9}$$

where $V'$ denotes the set of negative samples for $a_0$ and $\eta$ adjusts the weight of the contrastive term. It's important to note that, instead of iterating over all the $V'$ and $A'$ samples, we choose to randomly draw a subset from them per gradient descent step to reduce GPU memory consumption.

**Creating Negative Samples** In absence of a pre-existing high-quality dataset for contrastive learning, we can generate negative samples through data augmentation. Specifically, we employ the following methods to create $V'$ and $A'$ in the context of paired positive data $a, v$. For brevity, we will only outline the generation of negative audio samples $a^-$. The creation of negative videos $v^-$ follows a similar formulation:

- *Random Temporal Shifts*: We apply random temporal shifts to $a$, moving the signal backward or forward by a random duration within some hundreds of milliseconds.

---

**Algorithm 1** Training the joint diffusion model with a contrastive loss. $\mathcal{D}$ denotes the training dataset.

---

**repeat**
    $v_0, a_0 \sim \mathcal{D}$
    Create $V'$ and $A'$ with $\mathcal{D}$ and $v_0, a_0$
    $v_0^\neg, a_0^\neg \sim V', A'$ # randomly draw multiple negative samples and concat them as a batch
    $n \sim \mathcal{U}(1, 2, .., N)$
    $(\epsilon_a, \epsilon_a^\neg, \epsilon_v, \epsilon_v^\neg) \sim \mathcal{N}(\mathbf{0}, \mathbf{I})$
    $(a_n, v_n) = \sqrt{\alpha_n}(a_0, v_0) + \sqrt{1 - \alpha_n}(\epsilon_a, \epsilon_v)$
    $(a_n^\neg, v_n^\neg) = \sqrt{\alpha_n}(a_0^\neg, v_0^\neg) + \sqrt{1 - \alpha_n}(\epsilon_a^\neg, \epsilon_v^\neg)$
    $L = ||\mathbf{v} - \mathbf{v}_\theta(v_n, a_n, n)||^2$
    $L^\neg = ||\mathbf{v} - \mathbf{v}_\theta(v_n^\neg, a_n, n)||^2 + ||\mathbf{v} - \mathbf{v}_\theta(v_n, a_n^\neg, n)||^2$
    $L_{\text{cont}} = L - \eta L^\neg$
    $\theta = \theta - \varepsilon \nabla_\theta L_{\text{cont}}$ (learning rate: $\varepsilon$)
**until** converge

---

- *Random Segmentation and Swapping*: We randomly draw a separate audio segment, denoted as $a_d$, with the same length as $a$. Subsequently, we sample a random split point on both $a_d$ and $a$, allowing us to construct $a^\neg$ as either concatenate($a_d^{\text{left}}, a^{\text{right}}$) or concatenate($a^{\text{left}}, a_d^{\text{right}}$).

- *Random Swapping*: In this method, we randomly select a different audio segment, $a_d$, of the same length as $a$, and substitute $a$ with $a_d$.

The detailed training procedure is outlined in Algorithm 1.

## 4 Experiments

This section details the comprehensive evaluation of our Contrastive Multi-Modal Diffusion (CMMD) model, which we conducted using both subjective and objective measures on two distinct datasets. Furthermore, we demonstrate the speedup of our model resulting from its more efficient design. Additional details and results will be available in supplemental material.

**Datasets** Our evaluation leverages two datasets, each offering unique challenges and scenarios within the audio-video domain: **AIST++** (Li et al., 2021) is derived from the AIST Dance Database (Tsuchida et al., 2019). This dataset features street dance videos with accompanying music. It serves a dual purpose in our evaluation, being used for both video-to-audio and audio-to-video tasks. The **EPIC-Sound** (Huh et al., 2023) dataset consists of first-person view video recordings that capture a variety of kitchen activities, such as cooking, that are characterized by a strong audio-visual correlation. Due to the significant motion and camera movement in the videos, which complicates visual learning, we use EPIC-Sound exclusively for video-to-audio evaluation. We deliberately exclude other widely-used datasets such as the landscape recordings used in Ruan et al. (2023), as they exhibit very weak audio-visual synchronization, making evaluation impossible. However, we also note that our model still works well in unconditional generation task on landscape dataset and some qualitative examples are provided in Appendix A

**Baselines** The MM-Diffusion model (Ruan et al., 2023) stands as the only known baseline capable of handling both video-to-audio and audio-to-video synthesis tasks. For our comparison, we employed the official MM-Diffusion implementation, utilizing weights trained on the 1.6 s 10fps AIST++ dataset at a resolution of $64 \times 64$. Additionally, we present results from nCMMD, a variant of our CMMD model that does not incorporate contrastive loss.

**Feature Extraction & Data Preprocessing** We sampled 18 frames from 10 fps video sequences and the corresponding 1.8s audio at 16kHz. Video frames underwent center cropping and resizing to a $128 \times 128$ resolution, or optionally downsampling to $64 \times 64$ for a comparison with the MM-Diffusion baseline. The

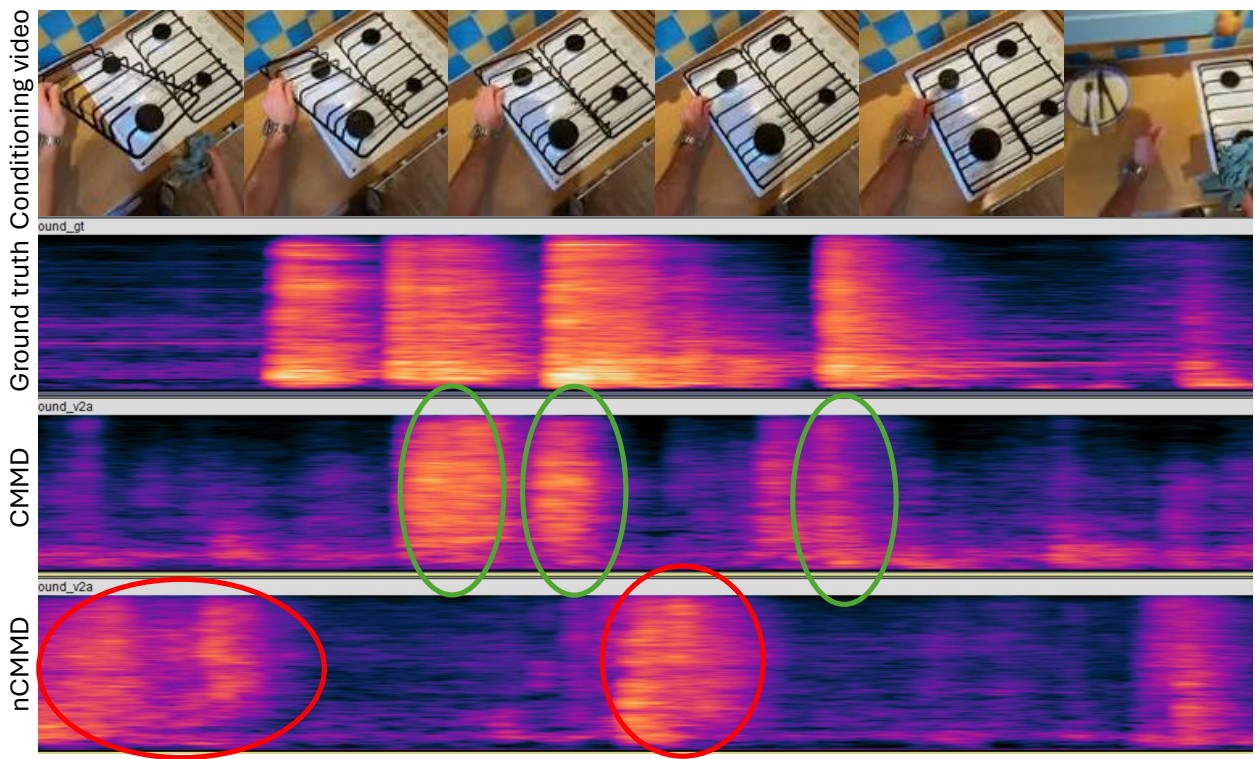

Figure 3: Conditioning video (top) with ground truth spectrogram below. The two bottom spectrograms show the generated audio with CMMD and nCMMD conditioned on the video. Sound events are highlighted with a green circle for matches and a red circle for mismatches.

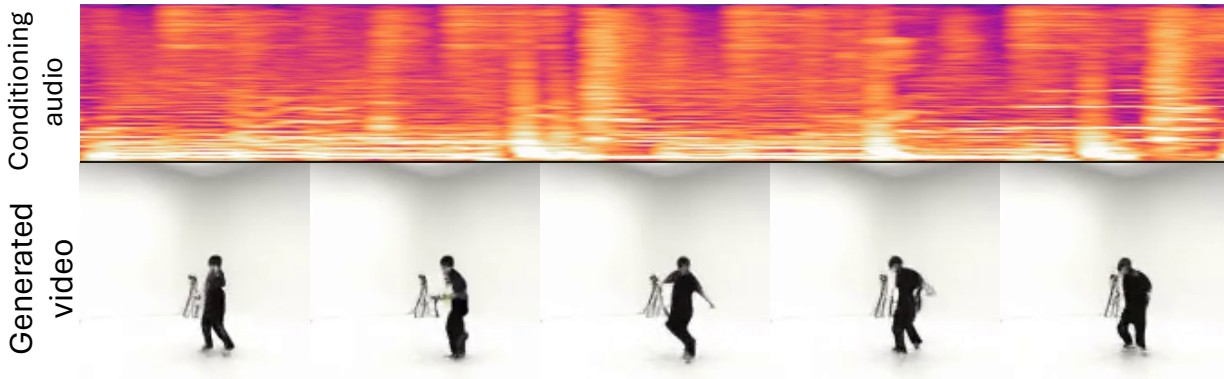

Figure 4: Generated video with CMMD conditioned on the audio shown as spectrogram above.

audio samples represented in a Mel Spectrogram have 80 channels and 112 time steps. During test time, we use twice the training sequence length, i.e., 36 video frames, if not specified otherwise.

As outlined in Section 3.2, we encoded videos using the Gaussian VAE from the Stable Diffusion project (Rombach et al., 2022), which effectively reduces image resolution by a factor of eight in both width and height. We utilized the pre-trained model weights[1] without further fine-tuning.

For audio features, we transformed waveforms sampled at 16 kHz into 80-bin mel spectrograms using a Short-Time Fourier Transform (STFT) with a 32 ms window and 50% overlap, yielding a time resolution

---

[1]https://huggingface.co/stabilityai/sd-vae-ft-mse

| Model | time | #params | video dim | latent dim | ms/FE |
|---|---|---|---|---|---|
| (n)CMMD | 1.8s | 106M | $128 \times 128$ | $16 \times 16$ | 136 |
| (n)CMMD | 1.8s | 106M | $512 \times 512$ | $64 \times 64$ | 299 |
| MM-Diff (Ruan et al., 2023) | 1.6s | 133M | $64 \times 64$ | - | 721 |

Table 1: Comparison of model size and computational complexity, where ms/FE represents milliseconds per function evaluation; (n)CMMD operates on a downsampled latent representation.

of 16 ms. The MelGAN vocoder was improved by the loss weightings from Hifi-GAN (Kong et al., 2020a) and notably improved by training on sequences of 4 s, as opposed to the originally suggested 0.5 s. This adjustment aligns with the MelGAN architecture's receptive field of approximately 1.6 s. The vocoder was trained on the entire AudioSet (Gemmeke et al., 2017) to ensure a broad sound reconstruction capability.

**Model Hyperparameters** Our nCMMD model was trained over 700,000 gradient steps with a batch size of 8 for versions excluding contrastive loss. For the full-fledged CMMD model, we began fine-tuning from a checkpoint at 400,000 steps of the nCMMD with $\eta = 5 \times 10^{-5}$ suggested by CDCD (Zhu et al., 2023), continuing until 700,000 steps with a reduced batch size of 2. The Adam optimizer was employed with an initial learning rate of $1 \times 10^{-4}$, which was annealed with a factor of 0.8 every $80,000$ steps until $2 \times 10^{-5}$. To create contrastive samples, we applied random shifts of $2 - 4$ frames (equivalent to $200 - 400$ ms) to either video or audio. We employ a cosine variance schedule for $\alpha_n$, and utilize 200 DDIM sampling steps for conditional generation.

## 4.1 Model Efficiency and Size

Before generative performance evaluation, we present a comparative evaluation of the inference efficiency between different backbone U-Net models on RTX Titan, as summarized in Table 1. To ensure a fair comparison, both models were executed on a $64 \times 64$ resolution space with a batch size of one. Notably, for (n)CMMD, this $64^2$-dimensional latent space is equivalent to operating on a $512 \times 512$ pixel space. For MM-Diffusion, we activated gradient caching for the best performance. Our evaluation involved a series of 100 denoising steps applied to video-to-audio generation tasks, from which we derived the average runtime. Additionally, the (n)CMMD model processed sequences of 1.8 s in length, compared to the 1.6 s sequences used by MM-Diffusion. Despite handling longer sequences, our model demonstrated a significant performance advantage, operating more than twice as fast and requiring 20% fewer parameters than the baseline model.

## 4.2 Metrics

**Fréchet Distance** Objective metrics to capture the perceived quality of video and audio are often difficult to develop and have many imperfections. Especially in generative tasks, where new content is created and no ground truth is available, such metrics are to be used with care. Popular approaches are statistical metrics, which compare generated and reference distributions in some embedding space, such as the *Fréchet Audio Distance (FAD)* (Kilgour et al., 2019) and *Fréchet Video Distance (FVD)* (Unterthiner et al., 2018). We assess FVD in a pairwise manner (Yang et al., 2022; Voleti et al., 2022): calculating the score between the 5 times conditional generation results and the corresponding ground truth test sets. To measure audio quality, we calculate FAD using CLAP embeddings (Elizalde et al., 2023), which have been shown recently in (Gui et al., 2024) to represent acoustic quality much better than the widely used VGGish features. FAD scores are calculated using the FAD toolkit (Gui et al., 2024) both individually for each generated sample and for the entire set of samples generated by one model, using the test set as a reference. Additionally, we also consider KVD (Bińkowski et al., 2018) as a complementary metric of visual quality for video contents.

**Temporal Alignment** For the dancing videos from AIST++, to evaluate the temporal alignment of generated music, we use a beat tracking approach similarly as in (Zhu et al., 2023) to measure the rhythmic synchronicity. The music beats are estimated using librosa (McFee et al., 2023) beat tracker and the hit rate between beats of generated and ground truth audio is computed. We propose to use a tolerance of $\pm 100$ ms, which corresponds approximately to the average perceivable audio-visual synchronicity thresholds

| Models | CMMD | | nCMMD | | MM-Diff | |
|---|---|---|---|---|---|---|
| | FVD | KVD | FVD | KVD | FVD | KVD |
| 16 frames (64) | **611** | 58 | 703 | 83 | 726 | **48** |
| 18 frames (64) | **749** | **70** | 799 | 187 | 757 | 71 |
| 32 frames (64) | 765 | 53 | **708** | **47** | 871 | 68 |
| 18 frames (128) | **934** | 78 | 1036 | 136 | N/A | |
| 36 frames (128) | 973 | **49** | **882** | 49 | N/A | |

Table 2: FVD and KVD results for different frame settings on AIST++ dataset. Numbers in parentheses indicate the resolution of the evaluated frames.

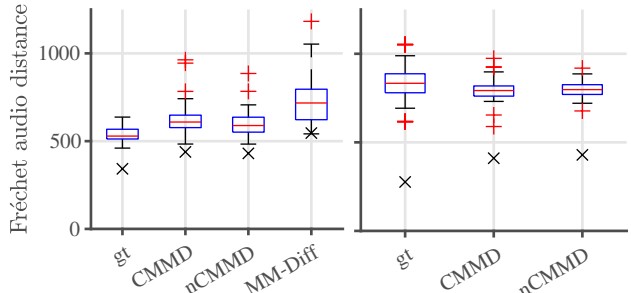

Figure 5: Per-sample (boxes) and per-set (×) Frechet audio distance (FAD) results for AIST++ (left) and EPIC-Sound (right). FAD is calculated for 50 output samples of each model using CLAP embeddings with the respective test set as reference. Boxes show the per-sample FAD distribution of these 50 samples, with red markers indicating outliers beyond the whiskers which extend to 1.5 times the interquartile range. Note that the per-set FAD scores for ground truth (gt) are larger than zero as only the small subset of the test set used in the evaluation is compared to the whole test set used as reference. Comparing FAD scores for identical set sizes avoids sample size bias (Gui et al., 2024).

found in literature (Younkin & Corriveau, 2008). For reference, we also show results using a larger tolerance of ±500 ms,which is equivalent to the 1 s quantization used in Zhu et al. (2023). While this significantly improve accuracy numbers, we consider this a very inaccurate, close to random metric, as a 1 s window already contains 2 beats at a average song tempo of 120 beats per minute. Since the beat tracking method is applicable only to musical content, we reserve the alignment assessment for EPIC-Sound to subjective evaluation.

**Subjective Evaluation** We conducted a user study with 14 participants to evaluate the audio-visual quality and synchronicity. Participants were recruited lab internally on voluntary basis without restrictions except unimpaired vision and hearing. No further demographic information was collected for privacy reasons. For each example, we asked two or three questions about the quality of the generated content and the temporal alignment of video and audio events on MOS scales from 1 (worst) to 5 (best). Specifically, for *generated video*, we asked to rate the *video quality* and the *temporal alignment*. For *audio generation* from AIST++ dance videos, we asked to rate separately the *acoustic* and *musical quality*, and the *temporal synchronization* of the dancer to the music. For the EPIC-Sound cases, we asked to rate the *acoustic* and *semantic quality*, and the *temporal synchronization* of events. Semantic quality refers to whether the type of sounds heard make sense given the scene seen in the video without paying attention to temporal synchronization.

### 4.3 Objective Evaluation Results

The results for Fréchet Video Distance (FVD) and Kernel Video Distance (KVD) comparing the proposed model and baseline models are detailed in Table 2. The findings reveal that (n)CMMD consistently outperforms MM-Diffusion across a variety of resolutions and sequence lengths. Specifically, CMMD demonstrates

| Hitrate tolerance | CMMD | nCMMD | MM-Diff (Ruan2023) | comments |
|---|---|---|---|---|
| ±500 ms | 89% | 91% | 89% | not suggested tolerance as in prior work |
| ±100 ms | **45%** | 44% | 41% | |

Table 3: Comparison of Beat Tracking Accuracy (AIST++). The values in parentheses indicate the allowable margin of error for beat timing, with a smaller window representing a stricter standard. Higher hit rates within lower tolerance thresholds signify superior temporal alignment.

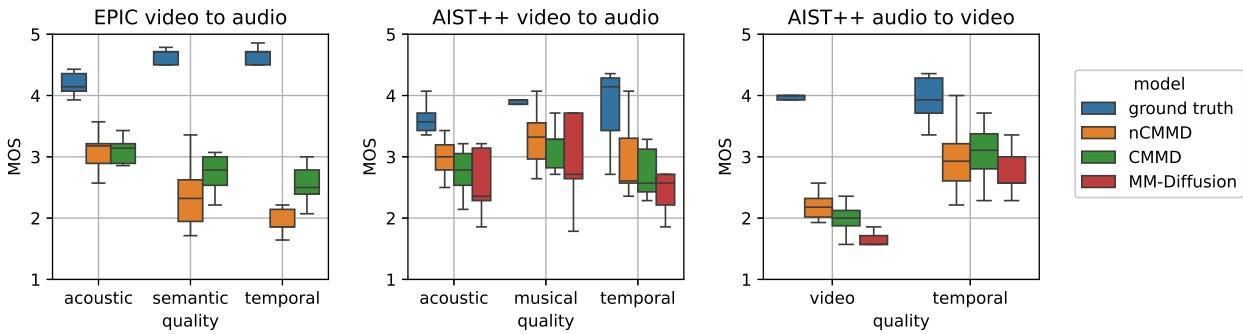

Figure 6: Subjective results from user study for EPIC-Sound video conditioned audio generation (left), AIST++ dance video conditioned audio generation (center), and audio conditioned video generation (right).

a marginal superiority over nCMMD in shorter sequences. Conversely, nCMMD exhibits slightly better quality in longer sequences, aligning with our subjective assessments. MM-Diffusion, however, performs better only in terms of KVD for low-resolution, short video sequences, which is the specific condition under which this model was trained.

Fig. 5 illustrates the comparison of audio quality in a video-to-audio generation scenario. Our CMMD model surpasses the baseline in AIST++ music audio quality, in terms of both per-sample FAD (Gui et al., 2024) and batch FAD metrics. There is no significant difference between CMMD and nCMMD for both datasets.

Table 3 presents the beat alignment results for the AIST++ audios. The table compares three different methods: CMMD, nCMMD, and MM-Diffusion. In terms of beat tracking accuracy within a 100 ms tolerance, CMMD performs the best, showing a improvement of 1-4%. As mentioned in 4.2, we do not consider the results for tolerance of 500 ms meaningful as it allows very coarse and ambiguous beat matches, so we do not suggest to draw conclusions from this setting. It is only notable that this large inaccurate tolerance doubles accuracy numbers, which we find misleading.

In the generation processes of audios for EPIC-Sound and videos for AIST++, our primary reliance is on subjective evaluation, given the absence of robust metrics. To supplement this assessment, we present EPIC-Sound audio generation visualization provided in Fig. 3, where we can observe that CMMD has better alignment with the ground truth than nCMMD in terms of temporal sound event alignment. Additionally, Fig. 4 presents a qualitative sample showcasing audio-to-video generation in the context of AIST++.

## 4.4 Subjective Evaluation Results

In the subjective evaluation we used 85 videos. We used 5 different conditions (audio or video as conditioning), two different sample generations per CMMD and nCMMD model, one sample each per ground truth and baseline. For AIST++ we evaluated audio to video and video to audio generation. For EPIC-Sound, we evaluated only video to audio and there is no MM-Diffusion baseline available.

The Mean Opinion Scores (MOS) are shown as boxplots in Fig. 6, where the black bars show then median, the boxes show the inter-quartile range, and the whiskers show the minimum and maximum values. Additionally, we test statistical significance using the Wilcoxon signed-rank test with p-value < 0.05 to analyze close

cases. We can see that the raters reliably detected the ground truth samples attributing it the highest score, although often the scale was not used fully. For the generated dance visuals from AIST++ audio (Fig. 6 right), we can observe a significantly higher rating of our proposed models over MM-Diffusion baseline. The nCMMD model has a slightly higher video quality with p=0.005. The CMMD model shows a trending but non-significant better temporal alignment than nCMMD with $p = 0.327$. CMMD temporal alignment is significantly better than MM-Diffusion baseline with $p = 0.038$.

For audio generation conditioned on AIST++ dance videos (Fig. 6 center), we observe the temporal alignment of CMMD and nCMMD as well as their acoustic quality better than MM-Diffusion. While these differences are smaller, they are statistically significant. nCMMD has slightly better acoustic quality than CMMD, while there is no significance between CMMD and nCMMD temporal alignment with $p = 0.09$, as can also be seen on their very close medians. nCMMD outperforms MM-Diffusion in musical quality, while the spread is too large to draw conclusions between CMMD and MM-Diffusion on musical quality.

For audio generation conditioned on EPIC-Sound videos (Fig. 6 left), CMMD significantly outperforms nCMMD in terms of semantic quality and temporal alignment due to the use of the contrastive loss, while the acoustic quality is on par.

### 4.5 Discussion

The results in Fig. 6 on EPIC-Sound video to audio task show a clear benefit of the contrastive loss to enforce stronger both temporal synchronization and semantic alignment without sacrificing audio acoustic quality. In AIST++, the contrastive loss improves the temporal synchronization MOS for the audio to video condition, while it is inconclusive for the video to audio condition. However, the $100ms$ beat tracking metric in Tab. 3 still indicates a minor synchronization improvement also for this the latter data condition. Interestingly, on AIST++ it seems that the model trades off a small amount of quality in favor of better synchronization, while objective metrics like FVD, KVD and FAD are on par or fluctuating depending on condition. In general, the temporal synchronization results are less pronounced for the AIST++ dance data, possibly due to the fact that the alignment of human dancers with music may be harder to judge for several reasons: 1) the dancers may vary in tempo or their internal rhythm may be judged in ambiguous ways. 2) being off by one or two full beats may appear as being in sync again.

## 5 Conclusion

The multi-modal diffusion model we presented marks an advancement in the field of bi-directional conditional generation of video and audio. Introduction of a more effective and efficient design and the joint contrastive loss were shown to be beneficial improvements. Our experiments across various datasets validate our model's superior performance over an existing baseline. The model not only excels in the quality of generation but also shows advantage in alignment performance, especially in scenarios demanding tight audio-visual correlation. However, our results indicate that there may be an inherent trade-off between generation quality and temporal alignment, which can be addressed in future work to analyze further and potentially improve. This research paves the way for future innovations in video and audio conditional generation. Moving forward, the model can serve as a foundational architecture for subsequent developments aimed at further refining the quality and alignment of generated video-audio contents.

**Broader Impact on Society on Risks** Our model's capabilities also introduce several risks that must be carefully considered. One of the primary concerns is the potential for misuse in creating deepfakes, which are increasingly realistic and difficult to detect. These can be used to spread misinformation, manipulate public opinion, or impersonate individuals, thereby undermining trust in media and harming reputations.

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

# A    Additional Landscape examples

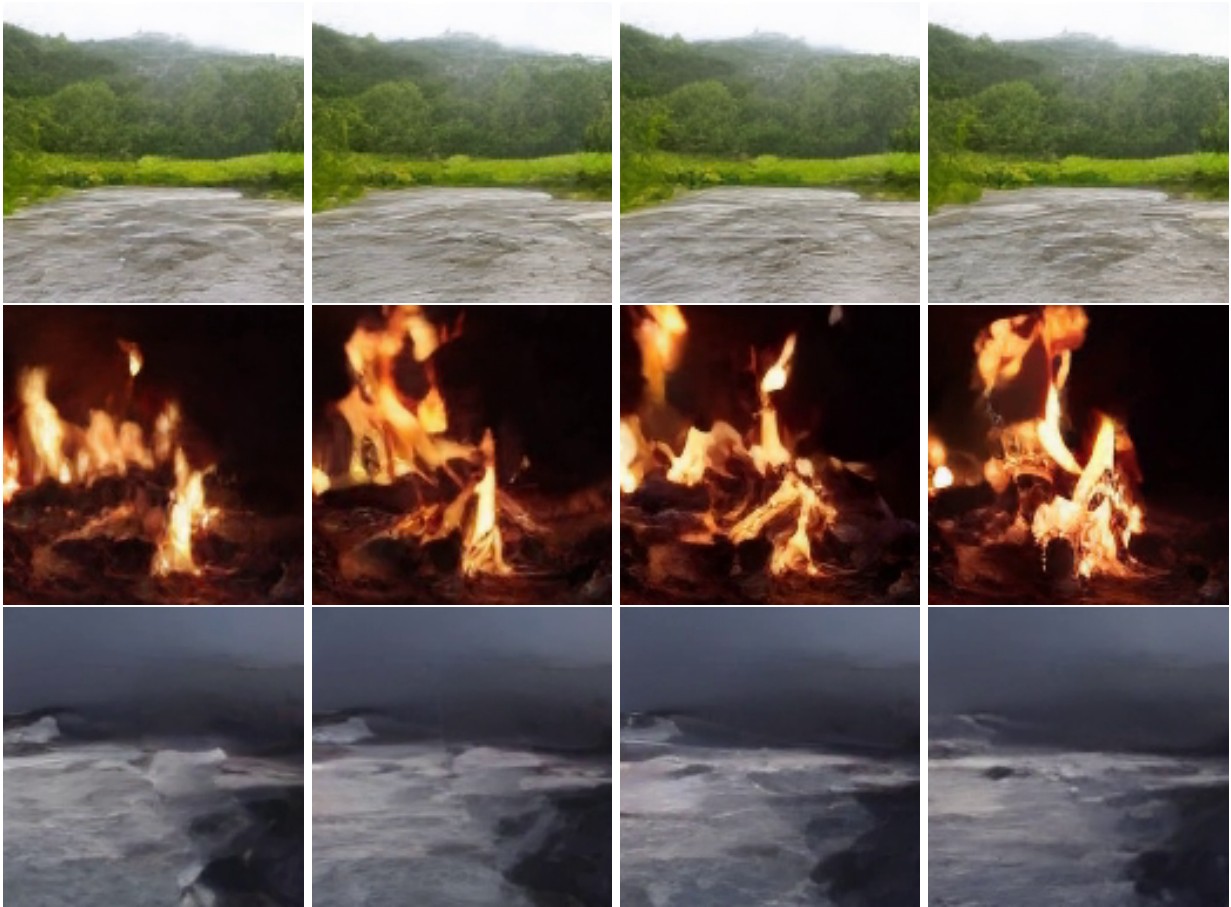

Figure 7: Unconditional generation samples from Landscape dataset.

# B    Additional Details On Model Structure

Table 4 shows details of the diffusion U-net architecture and training details.

| Configurable Components | Diffusion U-Net |
| --- | --- |
| Base channel | 128 |
| Channel scale multiply | 1,2,3,4 |
| Video downsample scale | H/2, W/2 |
| Audio downsample scale | T/2 |
| Attention dimension | 64 |
| Attention heads | Channel // Attention Dim |
| Diffusion noise schedule | cosine |
| Diffusion steps | 1000 for training |
| Prediction target | $\mathbf{v}$ |
| Sample method | DDIM |
| Sample step | 200 |
| $\eta$ | 5 |

Table 4: Supplemental Diffusion U-Net configuration details.

### B.1 MelGAN vocoder

We used the official *MelGAN* architecture[2] (Kumar et al., 2019) with only modifications in training procedure and loss weightings. We found that the proposed architecture form HiFi-GAN (Kong et al., 2020a) did not work better, but the loss weights provide a significant improvement. We noticed that the receptive field of the convolutional net is around 1.6 s, but training sequences are proposed to 8192 samples, which is even less than 0.5 s at the used sampling rate of 22.5 kHz. We noticed a large performance improvement by simply increasing our training sequence lengths to 4 s.

We trained on full Audioset data ($\sim$5000 h) in 16 kHz and augmentation by random biquad filtering to ensure generalization to arbitrary sounds.

| Configurable Components | MelGAN |
|---|---|
| Mel bins | 80 |
| Sampling rate | 16 kHz |
| Trainable parameters | 4.27 M |
| Training sequence length | 65536 samples |
| Feature loss weight | 2 |
| Mel spectrum loss weight | 10 |

Table 5: MelGAN Configuration

## C   Subjective Evaluation Details

In this section, we show the set of questions employed in our subjective evaluation. Participants are tasked with assigning a numerical rating on a scale ranging from 1(worst) to 5(best) for each question.

**AIST++ video-to-audio generation.**

1. Please rate the **acoustic quality** of the music, **independent of the visual aspect**.
   - Low quality might be pure noise, heavily distorted sound or non-musical audio.
   - Penalize any acoustic degradations like distortions, thin or muffled sound or other artifacts.
   - High quality is a good sounding dance music without severe artifacts.

2. Please rate the **musical quality**. Consider the following attributes in your judgement:
   - Does the music have a fluid rhythm or does the beat change randomly?
   - Does the music have a melodical theme or does it not sound appealing?

3. Please rate the **temporal synchronization** between dancer and the music.
   - Do the movements of the dancer seem to fit and be in sync with the music?

**AIST++ audio-to-video generation.**

1. Please rate the **quality** of the **video, independent of the sound**.
   - Very low quality might be very blurry or unrecognizable content.
   - Penalize artifacts like strange appearing body parts, separating bodies, physically impossible movements, etc.
   - High quality is a naturally looking video of a dancer. Required to answer. Single choice.

2. Please rate the **temporal synchronization** between dancer and the music: Do the movements of the dancer seem to fit and be in sync with the music?

---

[2]https://github.com/descriptinc/melgan-neurips

**EPIC-Sound video-to-audio generation.**

1. Please rate the **quality** of the sound, **independent of the visual aspect**.

   - Low quality might be pure noise or heavily distorted sound.
   - Penalize any acoustic degradations like distortions, thin or muffled sound or other artifacts.
   - High quality is a naturally and realistic sounding recording with a POV camera.

2. Please rate the **semantic relevance of audio, independent of the visual temporal alignment**:

   - Could the audio you hear appear in the environment you see (something out of the field of view could make this sound)?
   - Penalize sounds that would not appear in this kitchen scene.
   - Give a high rating, if the sounds you hear could appear in this kitchen scene, regardless of the video.

3. Please rate the **temporal correlation** between the audio and video events.

   - Do the audio events you hear occur at the same time as in the video?

# D   Additional Qualitative Examples

We provide video files in .mp4 format as supplementary qualitative samples. Please utilize any video player to evaluate the attached videos.

