# OpenReview forum: "CMMD: Contrastive Multi-Modal Diffusion for Video-Audio Conditional Modeling"
_TMLR — Rejected by TMLR_

### Review · Reviewer_hFpJ · 2024-04-11

**Summary Of Contributions:**

The paper aims to enhance the synchronization between video and audio generations using a contrastive loss function within a multi-modal diffusion model framework. It proposes an integrated approach with a pre-trained video autoencoder and a vocoder to improve the quality and alignment of video-audio pairs.

**Audience:**

Yes

**Broader Impact Concerns:**

The generated videos may be used for misinformation and other malicious use cases. It requires further discussion to address concerns about how the proposed method can avoid any harmful effects on society.

**Claims And Evidence:**

No

**Requested Changes:**

1. Evaluate the CMMD model on a more diverse dataset, such as the Landscape dataset, to compare with MM-diffusion and demonstrate its generalizability.
2. Include KVD metrics to assess the visual diversity of the generated content.
3. Provide complete details of the subjective evaluation methodology, including the demographic information of the raters, recruitment process, instructions, compensation, and compliance with minimum wage standards.
4. Address the potential loss of modality-specific information when enforcing synchronization.

**Strengths And Weaknesses:**

Strengths:
1. Enhanced synchronization between video and audio through contrastive constraints.
2. Experiments demonstrate the model's potential for generating quality synchronized content over certain datasets.

Weaknesses:
1. Limited evaluation datasets, missing out on the demonstrated diversity in MM-diffusion, such as Landspace, which is a real-world dataset.
2. The kernel video distance (KVD) metric as used in MM-diffusion, important for assessing visual diversity, was not considered.
3. Details about the subjective evaluation methodology are missing, notably the lack of demographic information on raters, such as their age, gender, ethnicity, and recruitment process, which affects the assessment's reliability.
4. The novelty of the methodology is questionable as it's a minor iteration (merely adding a well-known contrastive constraint) over existing frameworks.
5. The model potentially compromises unique modality information in favor of synchronization.

---

> ### Author Response · Authors · 2024-05-14
> **We thank the reviewer for the valuable feedback**
>
> We made several updates to the manuscript and marked major ones in blue font.
>
> - We agree the datasets are limited, but these were the only ones fitting our constraints of being available while having strong audio-visual temporal correlation. We trained our model on the Landscape data but realized it is near impossible to judge temporal synchronization of ocean waves, wind, fire etc. and therefore opted to not show results on this dataset. We will still provide unconditional examples for the Landscape dataset to the reviewers to demonstrate this case. (See Appendix A for some qualitative results.)
> - We added an additional KVD evaluation with more video samples.
> - We added information about the study participants, which were lab internal on voluntary basis. We did not collect any demographic information for privacy reasons and we did not deem it necessary for this study. Please note that the questions used in our user survey were already provided in the appendix.
> - The model builds on prior work, but we believe the proposed components such as multi-modal contrastive loss and Easy Fusion have sufficient novelty. We note that most of previous works focus on uni-modal conditional generation rather than bi-directional conditional generation, which is the core difference to most of the previous works.
> - Our results indicate indeed that modal quality may be traded off against synchronization, however not significant. We find this to be a valuable result to share and something future work can address. We added a comment in the conclusion for future work.
> - We also added a section to explain the potential broader impact.

---

### Review · Reviewer_WXhM · 2024-04-23

**Summary Of Contributions:**

This work considers the problem of learning to bi-directionally generate video from audio and vice versa using a machine learning model. The authors propose a modified U-Net-based architecture for joint audio-and-video generation, which combines pretrained encoder/decoder blocks with a novel "Easy Fusion" operation. They also propose a contrastive training objective, which constructs negative samples by perturbing the ground-truth targets, and which is intended to encourage better temporal synchronization. The authors show how their joint training objective can be used to perform conditional sampling of either video from audio or audio from video. On the AIST++ dataset, the authors demonstrate that their model outperforms the previously-proposed MM-diffusion model on the video-to-audio direction, and that there are some benefits to the contrastive loss (although the gains don't seem particularly consistent across evaluation methods). On the EPIC Sound dataset, the authors demonstrate that the contrastive loss improves the performance of their model relative to the non-contrastive variant.

**Audience:**

Yes

**Broader Impact Concerns:**

No broader impact concerns.

**Claims And Evidence:**

Yes

**Requested Changes:**

Before recomending the paper for acceptance, I think the authors should:

- Either (1) add more experimental comparisons to baselines to better support their claims about the efficacy of the proposed method or (2) decrease the strength of these claims throughout the paper,
- Make it more clear in the discussion in sections 4.5 and 5 that the temporal alignment benefits are only "clear" in a single task, not in all of the experiments,
- Fix the clarity issues (noted in more detail below).

I think the paper could also be strengthened by:

- Adding comparisons to audio-to-video and video-to-audio baselines for the relevant tasks, to get a better sense of how this work compares to previous approaches,
- Having at least one other task where there are baseline results, beyond AIST++ (this could be either EPIC Sound or a different dataset)
- Adding a background section to help readers who aren't already familiar with reconstruction guidance, MM-Diffusion, and other previous papers that the authors build on.

Specific clarity issues and typos that I noticed in the paper:

- Throughout the paper: Many citations of previous work are written in `\citet` style, when `\citep` style would be more appropriate. For instance, "Following breakthroughs in image (Ramesh et al. 2022) and audio generation (Liu et al. 2023)" instead of "Following breakthroughs in image Ramesh et al. (2022) and audio generation Liu et al. (2023)"
- End of section 2: The paper states that diffusion models have a "unique ability to model complex distributions", but that seems like a bold and controversial claim to make, and it also seems unnecessary. (Don't autoregressive transformer-based models also have a similarly strong ability to model complex distributions, especially when combined with techniques like VQ-VAE?)
- Equation 2 is missing a closing `]`
- In equation 5, I don't understand what "$v_n \sim q(v_n|v_0) \text{ with } \varepsilon_v$" means. What do you mean by "with $\varepsilon_v$"? More broadly, equation 5 could benefit a lot from a high-level explanation of what is being computed, instead of just writing the equations. (A background section, like I suggest above, could help here.)
- Section 3.3 states "M and $\eta$ denote the number of negative samples". Does that mean $M = \eta$? If so, why use two symbols? Also, what does the $\equiv$ symbol mean here? Are these exactly equal?
- Also in Section 3.3: "equation 5" is formatted strangely and not capitalized.
- Also in Section 3.3: The sentence "Given this deterministic relationship between $v_n$ and $v_0$, we can assume $p_\theta(a_n−1|a_n,v_0) \approx p_\theta(a_n−1|a_n,v_n)$ in equation 7" seems a bit hand-wavey to me. The deterministic relationship only holds when you condition on $\epsilon_v$, right? So the proper statement would be $p_\theta(a_n−1|a_n,v_0,\epsilon_v) \approx p_\theta(a_n−1|a_n,v_n,\epsilon_v)$, wouldn't it? Why is it OK to drop the $\epsilon_v$ dependence here?
- Also in Section 3.3: Both the original and swapped versions of the contrastive loss are referred to as the "video-to-audio" direction. I assume the second should be "audio-to-video"?
- In Algorithm 1, the comment says "draw multiple negative samples" but there is only one sample drawn. Are you actually averaging over multiple such samples? It might be clearer to show this with an explicit subscript/superscript.
- Start of Section 4: Typo "Muldi" instead of "Multi"
- "Model Hyperparameters" section: What does "exponentially annealed every 80,000 steps" mean?
- "Fréchet Distance" section: typo "manne" instead of "manner"
- Figure 5: It's hard for me to understand what this figure is showing, and I think the caption should go into more detail. Is this a box plot summarizing a distribution? This should be stated explicitly if so. Also, is it the distribution of many samples for the same conditioning signal, or of a single sample for each, or something else? How many samples were there? Are the red `+` symbols outliers? (Not necessary to change, but personally I prefer to avoid box plots for summarizing data, for many of the reasons described in [this page](https://nightingaledvs.com/ive-stopped-using-box-plots-should-you/).)

**Strengths And Weaknesses:**

### Strength 1: Interesting contributions in the architecture and loss
The proposed architecture and losses seem interesting and relevant to the community. I found the "easy fusion" in particular to be quite interesting; the authors are able to get many of the benefits of cross-attention by directly mixing activations and then applying ordinary self-attention.

### Strength 2: Both computational and human evaluations
The authors evaluate their approaches both in terms of computational metrics (such as Frechet Audio Distance / Frechet Video Distance) and in terms of subjective human judgements. This gives a good perspective on how the evaluated models perform.

### Weakness 1: Limited baselines and experimental settings
The authors emphasize that their approach, like MM-Diffusion, is multimodal and supports bi-directional inference, in contrast to a number of previous approaches that either consider just audio-to-video or just video-to-audio. However, I found the experimental setup to be fairly limited:

- Despite separately evaluating on audio-to-video and video-to-audio tasks, the only baseline the authors compare against on the AIST++ task is MM-Diffusion. It seems like it would be possible to compare directly to audio-to-video models for the audio-to-video task, and separately compare to video-to-audio models for the video-to-audio task.
- The other dataset the authors evaluate on, EPIC Sound, has no baselines at all. The authors are just comparing variants of their own approach.
- There is no evaluation of the joint generation ability of the model. It seems like one of the advantages of training an unconditional model is that it would support unconditional generation, but the authors don't seem to explore this.

By itself, this doesn't disqualify the paper from TMLR. But I think this means the conclusions we can draw from the paper are fairly weak, and I think the paper would be much stronger with more baselines. Also, the authors claim that their contribution includes "conducting comprehensive experiments on multiple datasets to thoroughly evaluate the efficacy of our proposed model", which feels like a stretch given that they are essentially comparing to a single baseline on a single dataset (and a single ablation on another dataset).

### Weakness 2: Multiple overstated/unjustified claims
The paper makes a number of claims that seem too strong based on the evidence provided.

- According to the experimental results for temporal alignment, the CMMD variant of their model significantly outperforms the nCMMD variant on *only one* of three tasks considered (the EPIC Sound task), and results for other tasks are inconclusive at best. However, the paper then states "The results on the EPIC-Sound and AIST++ audio to video show a very clear benefit of the contrastive loss to enforce stronger temporal synchronization and semantic alignment" and "CMMD significantly improves temporal synchronization". This seems too strong given the evidence provided.
- The introduction claims "We propose to use metrics with improved correlation human perception and practical relevance compared to prior work in the field, as we find several widely used metrics to have strong deficiencies." However, the paper doesn't seem to discuss any specific deficiencies that the authors find, and it seems like the authors are just re-using some existing metrics proposed by prior work.
- As mentioned in weakness 1, the paper introduction seems to exaggerate the thoroughness of the experiments.
- The paper says their contribution "marks an advancement in the field" which seems a bit grandiose.

### Weakness 3: Multiple distracting clarity issues

The paper has a number of clarity issues that made it harder to follow, and I still do not fully understand all parts of it. Some math expressions don't seem to be clearly defined, some figures were not fully explained, there are a number of citations using the wrong citation format, and I found multiple typos. I have listed these in more detail in the "Requested Changes" section below.

### Weakness 4: Not much background on prior approaches for unfamiliar readers.

I found some of the comparisons to prior work to be difficult to follow without previous familiarity with the cited works. For instance, the "Guided Conditional Generation" part of Section 3.1 just states a bunch of equations without explaining what "reconstruction guidance" is or how it relates to the proposed approach, and Section 3.2 says "the foundational structure draws some inspiration from MM-diffusion" but doesn't elaborate on what this inspiration was or on how the structures are similar. I think the paper could benefit from a background section that describes more clearly what previous ideas the proposed architecture is building on.

---

> ### Author Response · Authors · 2024-05-14
> **We thank the reviewer for the detailed and valuable feedback**
>
> We made several updates to the manuscript and marked major ones in blue font.
>
> - Our primary contribution lies in the synchronized bi-directional conditional generation of video and audio, which inherently differs from traditional uni-directional generative models, making direct comparisons less straightforward. For the EPIC-Sound dataset, we conducted an ablation study to evaluate the impact of the contrastive loss. Additionally, for demonstrating our model's capabilities in unconditional generation, we included qualitative video examples from the Landscape dataset in Appendix A. However, it is important to emphasize that our methodology is specifically designed for bi-directional conditional generation.
> - We have a background section (Sec.~2), provide references and short explanations where necessary, and improved the clarification for reconstruction guidance at equation (5). We believe this followed a good balance between self-contained clarity and not overly stretching the paper with explaining existing techniques. If there are any other specifics the reviewer would like us to clarify/elaborate, we are happily taking more suggestions into consideration.
> - The temporal synchronization has been rated subjectively better with clear statistical significance ($p \ll 0.05)$ for CMMD for both video scenarios in EPIC and AIST++ datasets. Furthermore, the beat-tracking metric shows also an improvement for the contrastive loss with the appropriately chosen beat window of 100ms. The 500ms window is only shown for comparison with prior literature, but we do not deem the metric with such a large error tolerance useful. We therefore would argue that the improvement given by the contrastive loss can be shown in more than one datasets and more than one metric.  Therefore we believe the results are sufficiently reliable and demonstrate that the design is following its purpose.
> - We toned down the overstated claims.
> - We addressed all clarity issues indicated by reviewers in the updated manuscript, the major ones summarized below:
>   - Formatting issue and typos were fixed
>   - Equation 5 is updated.
>   - We added a clarification how on the negative samples: We reduce the need to for M samples by using fewer negatives and increaseing their weight with the factor $\eta$.
>   - We made the conditional distribution in section 3.3 clearer.
>   - In section 3.3, Eq.7 shows the original video-to-audio contrastive loss; Eq.9 and Algorithm 1 shows our generalized contrastive loss for bi-directional generation, which considers both negative video and negative audio; the last paragraph demonstrate how we create negative audio samples. To create negative video samples, we simply follow the same method.
>   - We improved the hyper-parameter paragraph.
>   - We added the following explanation to Fig. 5 to clarify the boxplot: The box plot summarizes the distribution of sample-wise Fr\'echet Audio Distance (FAD), where each sample is the model output to one conditioning signal, and the FAD is calculated between each sample and all ground-truth audio samples in the corresponding test set (AIST++ or EPIC-Sound). Each box shows the FAD distribution of 50 outputs of one model (CMMD / nCMMD / MM-Diff). The red symbols are outliers, defined as FAD values outside the whiskers, which extend to 1.5 times the interquartile range. The x symbols indicate the FAD between the entire set of 50 model output samples and the ground-truth test set.

---

> > ### Comment · Reviewer_WXhM · 2024-05-16
> > **Discussion**
> >
> > Thanks for the updates. My responses to each point are below.
> >
> > **Bidirectional conditional generation evaluation:** I still think the paper would be stronger with more comparisons to previous work (e.g. comparing to MM-Diffusion on EPIC-Sound), but I understand that direct comparison is somewhat difficult if previous models did not evaluate on these datasets. (It still seems like you could in principle compare directly to unidirectional generative models, though? This would put your method at a disadvantage, I suppose, but if it still did well that would be impressive. Is there something I'm missing that makes this harder than it seems?)
> >
> > Thanks for adding the unconditional generation results for the Landscape data (although I notice Figure 7 just says "Caption" right now, which probably isn't intentional). Do I understand right that you don't directly compare on Landscape data because you don't trust quantitative evaluation methods for this task?
> >
> > **Background:** The current Section 2 seems to mostly contain references to papers without actually describing the technical details that this work builds on. By "background section", I meant that it would be useful to have a section that describes the relevant details instead of just pointing to the papers, so that readers do not have to refer to a different paper in order to understand how your approach builds on previous ones. (But I don't think this is strictly necessary.)
> >
> > A specific thing that I think would be useful to add is some additional discussion of how "the foundational structure draws some inspiration from MM-diffusion". For instance, you might say something like "Drawing inspiration from MM-diffusion, we {do X}. However, instead of {Y}, we do {Z}", or something like that. As written, I'm not sure what parts of the structure were inspired by MM-diffusion (is it just the U-net paradigm, or more than that?)
> >
> > Thank you for clarifying the reconstruction guidance explanation in Section 3.1, which I agree is clearer now.
> >
> > **Significance of empirical results:** I'm still finding it hard to understand what significant improvements you are referring to. From my read:
> >
> > - Objective results:
> >   - FVD and KVD are better for (n)CMMD than MM-Diff on AIST++, except 16 frames KVD. CMMD is not clearly better than nCMMD.
> >   - Beat alignment is better than MM-Diff at 100ms. CMMD is 1\% better than nCMMD, which is "not substantial" (according to the paper).
> > - Subjective results:
> >   - Significantly better ratings on AIST++ audio -> video for (n)CMMD over baseline, not significant between CMMD and nCMMD
> >   - Non-significant differences on AIST++ video -> audio for (n)CMMD over baseline
> >   - No baseline for EPIC-Sound video -> audio, but a significant difference between the two proposed methods CMMD and nCMMD.
> >
> > You say "The temporal synchronization has been rated subjectively better with clear statistical significance ($p \ll 0.05$) for CMMD for both video scenarios in EPIC and AIST++ datasets." But, unless I'm misunderstanding something, that doesn't seem to match the actual results. It seems like the paper says that CMMD is only significantly better than nCMMD on one task, EPIC-Sound video -> audio. (Although CMMD trends better than nCMMD on AIST++ audio-to-video temporal quality, CMMD seems to do slightly worse on AIST++ video-to-audio relative to nCMMD, and also on video quality for audio-to-video.)
> >
> > I agree that CMMD and nCMMD together seem to do better than MM-Diffusion on multiple tasks and metrics, but the specific impact of the contrastive loss (CMMD vs nCMMD) seems inconclusive from the results given here. Given this, I still feel like the claim "The results on the EPIC-Sound and AIST++ audio to video show a very clear benefit of the contrastive loss to enforce stronger temporal synchronization and semantic alignment." is not well supported by evidence.
> >
> > **Clarity issues:** Thanks for addressing many of my points. Some remaining issues:
> >
> > - Equation 7: You still haven't explained what $\equiv$ means here.
> > - Section 3.3: It still seems like there's an issue with the notation for the "swapped" version. You start by saying Equation 7 "integrates contrastive loss into the video-to-audio conditional diffusion models", and say that you can "bridge Eq.7 to our jointly trained multi-modal diffusion model", but then you say "For video-to-audio generation, we can follow the same method above by swapping $v$ and $a$". But this was already the video-to-audio generation. Shouldn't this be "For **audio-to-video** generation, we can follow the same method above by swapping $v$ and $a$"
> > - Algorithm 1: I still think this algorithm is unclear; it is written as if you draw one sample but it just says "draw multiple negative samples". The algorithm should actually show the steps taken; if you are drawing multiple samples you should write down multiple samples in the algorithm and show how you combine them (e.g. with a for loop or with an index and a summation).

---

> > > ### Author Response · Authors · 2024-05-18
> > > **Thank you**
> > >
> > > - Baseline:
> > >   - We appreciate the suggestion to include additional baseline comparisons to strengthen our paper. However, incorporating certain baselines, such as MM-Diffusion, presents significant challenges due to the computational resources required. Specifically, MM-Diffusion was trained on both pixel and waveform spaces using 32 V100 GPUs, as detailed in their [Appendix](https://arxiv.org/pdf/2212.09478). This level of resource allocation far exceeds what is feasible for our project, making such a comparison impractical.
> > >   - Regarding the inclusion of uni-conditional generation baselines such as video-to-audio models, it is important to consider the inherent advantages these models hold over bi-direction model. (There is always a trade-off between a general multi-purpose model and a specialized model)
> > >   - Models like Diff-Foley, as referenced in their [paper](https://arxiv.org/pdf/2306.17203), are specifically optimized for a single task and consequently have an unfair advantage over our multi-purpose model. Furthermore, even the smallest version of Diff-Foley has three times as many parameters as our model, leading to another layer of disparity in any direct comparison.
> > >   - Given these factors, we prioritize comparisons that are both feasible and fair, ensuring that our evaluations are meaningful and reflective of our model's capabilities within reasonable computational constraints.
> > > - Background
> > >   - Our section 2 is related works section, which is aimed to provide a very brief overview about the related previous works. The beginning of section 3 is more like a "background" section and it shows how our method is built upon diffusion model.
> > >   - As you suggested, we improve our description about how our method was inspired by MM-Diffusion.
> > > - Significancy
> > >   - The reviewer may think that we are claiming significant improvements in terms of **generation quality** for CMMD vs. nCMMD, which is never the case in the manuscript. We only observe significant improvements for the **metrics related to synchronization**, i.e. beat tracking, Epic video to audio (temporal), AIST audio-to-video (temporal). The differences AIST video-to-audio (temporal) are insignificant, therefore not conclusive. Note that the uni-modal metrics FAD, FVD, KVD do not measure cross-modal synchronization. We therefore count three positive vs. one non-conclusive metrics for better synchronziation using contrastive loss here. We furthermore do mention the possible trade-off between synchronization and quality as unexplored and left for future work in the conclusion.
> > > - Clarity:
> > >   - We think $\equiv$ should be $\approx$, because we only draw a subset of contrastive samples at each training step. See revised draft.
> > >   - You are correct, it should be "audio-to-video" and we revised the paper accordingly.
> > >   - We further add a sentence in algorithm 1. Hope it helps.

---

> > > > ### Comment · Reviewer_WXhM · 2024-05-21
> > > > **Continued discussion**
> > > >
> > > > Thanks for the updates and the clarifications.
> > > >
> > > > Regarding synchronization metrics: I'm still not convinced the metrics you present indicate "a very clear benefit of the contrastive loss to enforce stronger temporal synchronization and semantic alignment" even if you look just at the synchronization metrics. For these metrics:
> > > >
> > > > - Beat tracking at 500 ms: nCMMD beats CMMD by 2% (which you say is "not substantial")
> > > > - Beat tracking at 100 ms: CMMD beats nCMMD by 1% (which is smaller than the "not substantial" 2% drop above)
> > > > - EPIC-Sound video -> audio, subjective: CMMD beats nCMMD (significant)
> > > > - AIST++ video -> audio, subjective: nCMMD seems slightly above CMMD (the paper says this is "slight but insignificant
> > > > trend of better quality for the non-contrastive loss")
> > > > - AIST++ audio -> video, subjective: CMMD seems slightly above nCMMD (the paper says this is "a trending but
> > > > non-significant better temporal alignment")
> > > >
> > > > If a 2% drop in beat tracking accuracy (at 500 ms) is "not substantial", I would assume that a 1% improvement (at 100ms) would also be not substantial. And, for the subjective analysis, it seems like there's only one conclusive positive finding, with the other two insignificant. So we are left with one strong positive finding, two weak (insignificant or non-substantial) positive findings, and two weak (insignificant or non-substantial) negative findings. This doesn't seem like "a very clear benefit" to me?
> > > >
> > > > I'd be satisfied if you reworded Section 4.5 to be more specific about the claimed improvements. Instead of "The results on the EPIC-Sound and AIST++ audio to video show a very clear benefit of the contrastive loss to enforce stronger temporal synchronization and semantic alignment", you could change it to something like "For enforcing stronger temporal synchronization and semantic alignment, our results on the EPIC-Sound video-to-audio task indicate a clear benefit of the contrastive loss. Our proposed models also outperform the MM-diffusion baseline on AIST++ audio-to-video, although the impact of the contrastive loss itself is inconclusive for this task."
> > > >
> > > > Also, the claim "CMMD significantly improves temporal synchronization" later in that paragraph only seems true of the EPIC-Sound task, but this isn't explicitly stated. It would be good to clarify this.
> > > >
> > > > Separately, I notice that section 5 says "Our experiments across various datasets validate our model’s superior performance over existing baselines." Since you only compare a single baseline, MM-diffusion, it would be better to state this explicitly, since the statement currently implies there were multiple baselines.

---

> > > > > ### Author Response · Authors · 2024-05-23
> > > > > **Thank you**
> > > > >
> > > > > - We thank the reviewer for pointing out the additional inconsistencies. We noticed the subjective results were inaccurately described and we had used information from an outdated and buggy statistical significance test. We added more information on the statistical significance between the MOS measurements and added some p-values where necessary.
> > > > >   - The minor quality hits for AIST++ video and audio are still significant, so we clearly mention that now.
> > > > >   - The temporal alignment results between CMMD and nCMMD for EPIC and AIST++ audio to video are significant, while the AIST++ video to audio temporal alignment differences are non-significant.
> > > > >   - We added a clear note in the table and the text that we do not consider the +/-500ms tolerance a useful metric at all, we just added it for information and to compare numbers to prior literature.
> > > > > - Therefore, we would like to re-iterate, as mentioned in our prior response, that it is fair to say that we have 3 positive and one inconclusive metrics indicating that the contrastive loss yields better temporal alignment. I hope the reviewer agrees that this is sufficient to support the claim of improved temporal alignment, at the possible cost of other quality. We however also re-wrote the discussion section to reflect the results more distinct and took care to not over-claim anything.

---

> > > > > > ### Comment · Reviewer_WXhM · 2024-05-24
> > > > > > **Updated review**
> > > > > >
> > > > > > Thanks for the additional clarifications and for rephrasing the discussion to be more precise, I think it is clearer now which findings are significant and what the claimed improvements are.
> > > > > >
> > > > > > My important concerns with this work have now been addressed. I have thus changed my "Claims And Evidence" recommendation from "No" to "Yes".

---

### Review · Reviewer_Fs2i · 2024-04-30

**Summary Of Contributions:**

In general, this work bring the contrastive training into cross-modality and also studies some diffusion model based technique to synthesize video from input audio. The proposed CMMD model is to make the bi-directional conditional generation of video and audio content with better performance. The latent-spectrogram diffusion process reduces the dimensionality of the data, making the model more manageable and less resource-intensive without losing significant detail in the audio or video signals.

**Audience:**

Yes

**Broader Impact Concerns:**

No.

**Claims And Evidence:**

No

**Requested Changes:**

The current model and evaluation is mainly based on the well structured dataset.
It is possible to do some out-of-distribution or unseen generative on this audio-visual tasks.
I feel less confident to acknowledge the generalization and its zero-shot ability, where this task in the real world might be less under same source of data.

**Strengths And Weaknesses:**

**Pros / Sth New**
- new combination of dual U-Net structure and the joint contrastive diffusion loss
- empirical performance, in terms of synchronization between video and audio, which is a crucial aspect of multi-modal generation.
-  bi-directional generation of video and audio

**Cons**
- the complexity of the architecture might pose scalability issues, particularly for higher-resolution and longer-duration content
- I think it is a more systemic paper, there is no algorithmic novelty but it is ok

---

> ### Author Response · Authors · 2024-05-14
> **We thank the reviewer for your valuable feedback**
>
> We made several updates to the manuscript and marked major ones in blue font.
>
> We agree with the concern that our paper does not give insight on the generalization ability to large-scale data covering all imaginable cases. MM-Diffusion has provided examples on their Github page by training their unconditional model on the 2~M Audioset clips, showing a proof of concept that these models may generalize to larger data. To save energy resources and reduce impact on the environment (and due to the limit access to computational resources), we focused only on comparative improvements for constrained datasets. We furthermore did not have access to a large-scale training dataset with strong audio-visual correlation, which is not the case for Audioset and therefore disqualifies it for our purpose focusing on temporal synchronization.

---

### Decision · Action_Editor_D39D · 2024-07-02

**Recommendation:** Reject

**Comment:**

There was quite some discussion among one reviewer and the authors which led to several adjustments of the made claims (mainly toning down the claims and making them more precise). However, the claims are still not always well aligned with the results. E.g., the authors say "The findings reveal that (n)CMMD consistently outperforms MM-Diffusion across a variety of resolutions and sequence lengths." just to clarify this a few sentences later by "MM-Diffusion, however, performs better only in terms of KVD for low-resolution, short video sequences, which is the specific condition under which this model was trained." While this might be acceptable, the abstract still does not reflect this aspect, saying "Our findings demonstrate that the proposed model outperforms the baseline in terms of quality and generation speed through introduction of our novel cross-modal easy fusion architectural block.".

Considering the discussion of the reviewers and the authors and the still not well adjusted claims, I think the authors need to carefully adjust the claims and the experimental results before the paper should be accepted. Using further datasets and a more thorough statistical evaluation would also clarify whether the made statements hold more generally.

**Audience:**

Yes, the paper would be relevant to a some part of TMLRs audience as conditional generation of multi-modal data can be important in various domains.

**Claims And Evidence:**

The claims are only partly supported clearly by the provided evidence. Although the paper improved in this regard throughout the discussion with the reviewers, the clear picture sketched in the claims is not consistently supported by the presented results. For instance, the added KVD results are not conclusive and don't allow to draw conclusions about superior performance of the proposed approach. Also statistical evaluation is missing in this regard. Other valid concerns of the reviewers include limited benchmark datasets to justify making the claims in the stated generality ("We present experiments on multiple datasets..." - the paper considers two datasets).

**Resubmission Of Major Revision:**

The authors may consider submitting a major revision at a later time.